# Deep Optimal Transport: A Practical Algorithm for Photo-realistic Image Restoration

**Theo J. Adrai**
Technion–Israel Institute of Technology
Computer Science
theoad@cs.technion.ac.il

**Guy Ohayon**
Technion–Israel Institute of Technology
Computer Science
ohayonguy@cs.technion.ac.il

**Michael Elad**
Technion–Israel Institute of Technology
Computer Science
elad@cs.technion.ac.il

**Tomer Michaeli**
Technion–Israel Institute of Technology
Electrical Engineering
tomer.m@ee.technion.ac.il

## Abstract

We propose an image restoration algorithm that can control the perceptual quality and/or the mean square error (MSE) of any pre-trained model, trading one over the other at test time. Our algorithm is few-shot: Given about a dozen images restored by the model, it can significantly improve the perceptual quality and/or the MSE of the model for newly restored images without further training. Our approach is motivated by a recent theoretical result that links between the minimum MSE (MMSE) predictor and the predictor that minimizes the MSE under a perfect perceptual quality constraint. Specifically, it has been shown that the latter can be obtained by optimally transporting the output of the former, such that its distribution matches the source data. Thus, to improve the perceptual quality of a predictor that was originally trained to minimize MSE, we approximate the optimal transport by a linear transformation in the latent space of a variational auto-encoder, which we compute in closed-form using empirical means and covariances. Going beyond the theory, we find that applying the same procedure on models that were initially trained to achieve high perceptual quality, typically improves their perceptual quality even further. And by interpolating the results with the original output of the model, we can improve their MSE on the expense of perceptual quality. We illustrate our method on a variety of degradations applied to general content images of arbitrary dimensions.

## 1 Introduction

Many image restoration algorithms aim to recover a clean source image from its degraded version. The performance of such algorithms is often evaluated in terms of their average *distortion*, which measures the discrepancy between restored images and their corresponding clean sources, as well as *perceptual quality*, which refers to the extent to which restored images resemble natural images. The work in [2] exposed a fundamental trade-off between distortion and perceptual quality, where the latter is measured using a *perceptual index* that quantifies the statistical divergence between the distribution of restored images and the distribution of natural images. The

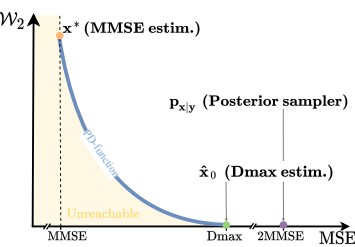

Figure 1: The $\mathcal{W}_2$-MSE trade-off [1].

37th Conference on Neural Information Processing Systems (NeurIPS 2023).

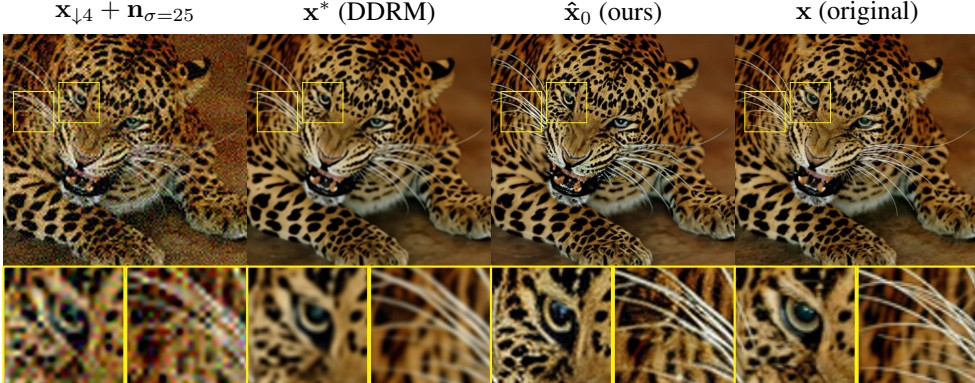

| $\mathbf{x}_{\downarrow 4} + \mathbf{n}_{\sigma=25}$ | $\mathbf{x}^*$ (DDRM) | $\hat{\mathbf{x}}_0$ (ours) | $\mathbf{x}$ (original) |

Figure 2: Our few-shot algorithm improves the visual quality of any estimator at test time. For example, we can improve the photo-realism of DDRM [3] even further.

trade-off curve reveals the predictor that achieves the lowest possible distortion, denoted as $\mathbf{D_{max}}$, while maintaining perfect perceptual quality (refer to fig. 1).

Following the methodology introduced by [2], it has become common practice to compare restoration methods on the perception-distortion (PD) plane, with many methods aiming to reach the elusive $\mathbf{D_{max}}$ point. In this paper, we present a practical approach to approximate the $\mathbf{D_{max}}$ predictor where distortion is measured using the MSE and perceptual quality is measured by the Wasserstein-2 distance ($\mathcal{W}_2$) between the distributions of restored and real images. Our approach is based on the recent work [1] which demonstrated that the $\mathbf{D_{max}}$ predictor can be obtained using *optimal transport* (OT) from the output distribution of the MMSE predictor to the distribution of natural images. By applying an optimal transport plan to an MMSE restoration resulting from a degraded image, we can produce $\mathbf{D_{max}}$ estimations by transporting the MMSE restored estimate.

Although progress has been made in finding OT plans between image distributions [4–6], it remains challenging task, particularly for high-dimensional distributions. Therefore, we propose an approximation method for the $\mathbf{D_{max}}$ estimator by performing transportation in the latent space of a pre-trained auto-encoder. A similar strategy was successfully employed in the context of image generation in [7], showing effectiveness in reducing complexity and preserving details.

Inspired by the style transfer literature [8–10], we assume that the latent representations follow a Multivariate Gaussian (MVG) distribution. Thus, by considering the first and second-order statistics of the embedded MMSE estimates and embedded natural images, we can compute the well-known closed form solution of the OT operator between two Gaussians. To further reduce complexity, we make additional assumptions about the structure of the latent covariance matrices, enabling the computation of the OT operator with as few as 10 *unpaired* MMSE restored and clean samples. This approach leads to a few-shot algorithm that significantly enhances visual quality.

Interestingly, our method can even improve the visual quality of generative models that were trained to achieve high perceptual quality in the first place (see fig. 2). Furthermore, by adjusting a single interpolation parameter, we can trade off perception for distortion, resulting in marginal improvements in the distortion performance of some regression models that were trained to prioritize source fidelity. We demonstrate the improved photo-realism of our approach on a variety of tasks and models, including GAN and diffusion-based methods, using high-resolution (*e.g.*, $512^2$ px) general-content images with arbitrary aspect ratios. [1]

## 2   Related Work

Throughout the paper, we distinguish between two kinds of restoration algorithms: distortion and perception focused. The former category includes traditional methods that minimize distortion (e.g.,

---

[1]Our code is publicly available at `https://github.com/theoad/dot-dmax`.

MSE) [11–15]. The latter category includes more recent works that usually involve generative models like Generative Adversarial Networks (GANs) [16–18], or diffusion-based techniques [3, 19].

This paper searches for the theoretical $\mathbf{D_{max}}$ estimator which minimizes the MSE under a perfect perceptual quality constraint. However, our method is not a stand-alone restoration algorithm, i.e., its input is not the degraded image. Rather, it can be applied on top of any *existing* predictor. We provide a new way to potentially improve the performance (either MSE or perceptual) of any given estimator in a few-shot, plug-and-play fashion.

Provided with an image latent representation method (e.g., an auto-encoder), our algorithm applies a linear transformation on all the overlapping patches of its input (after encoding). In this regard, its functioning is not far from classical image restoration methods [14, 15, 20].

## 2.1 Wasserstein-2 transport

While many successful approaches exist to compute the $\mathcal{W}_2$ distance between discrete, low or medium dimensional densities [21, 22], it is far more challenging to determine an optimal transport plan in the continuous, high-dimensional setting. In fact, the task of even computing the Wasserstein distance (without its optimal plan) on empirical distributions has drawn significant attention with WGANs [23]. Thus, computing the transport operator requires to optimize the $\mathcal{W}_2$ distance but with an additional ordering constraint on the generator, which proved to be a challenging task when dealing with real-world data sets [4–6]. An attempt to sidestep this difficulty would be to use the Gelbrich distance [1, 24], which lower bounds the $\mathcal{W}_2$ distance and depends only on the first two moments of the distributions. Nevertheless, it is only a good estimate when the support of the distributions has elliptical level sets. To address this, one can find a low-dimensional embedding where (i) high dimensionality is no longer an issue, (ii) the distributions are not degenerated, and (iii) the Gelbrich distance equals the Wasserstein-2 distance. A possible option would be to use the bottleneck of an auto-encoder. This approach was adopted by style transfer works [8–10], and is also widely used by tools that compare image distributions like the Fréchet Inception Distance (FID) [25]. Both use a convolutional encoder and consider the pixels of the latent embedding (the vectors that span across the channel dimension) as a MVG distribution.

# 3 Background

## 3.1 Optimal transport in Wasserstein Space

In this section, we briefly introduce key concepts of optimal transport theory that we draw from [26].

Let $\mu$ and $\nu$ be probability measures on $\mathbb{R}^m$ and $\mathbb{R}^n$, respectively. The set of all transport plans, which are probability measures $\pi$ on $\mathbb{R}^m \times \mathbb{R}^n$ with marginals $\mu$ and $\nu$, is denoted by $\Pi(\mu, \nu)$. The *Wasserstein-2 distance* between $\mu$ and $\nu$ is defined as follows:

$$\mathcal{W}_2^2(\mu, \nu) = \inf_{\pi \in \Pi(\mu, \nu)} \mathbb{E}_{x, y \sim \pi} \left[ \|x - y\|_2^2 \right]. \tag{1}$$

A transport plan that achieves this infimum is called an *optimal transport plan* between $\mu$ and $\nu$. If $\mu$ has a density (*i.e.* is absolutely continuous w.r.t. the Lesbegue measure), there exists a measurable function $\mathrm{T}_{\mu \longrightarrow \nu} : \mathbb{R}^m \longrightarrow \mathbb{R}^n$ such that, if $\mathbf{x}_1 \sim \mu$ and $\mathbf{x}_2 \sim \nu$ are two random variables, then $\mathbf{x}_2 \overset{a.s}{=} \mathrm{T}_{\mu \longrightarrow \nu}(\mathbf{x}_1)$. We refer to $\mathrm{T}_{\mu \longrightarrow \nu}$ as the *optimal transport operator* between $\mu$ and $\nu$. Like in [5], we also abuse this notation even when $\pi$ is non-degenerate, in which case $\mathrm{T}_{\mu \longrightarrow \nu}$ represents a one-to-many (stochastic) mapping.

Additionally, when considering two Multivariate Gaussians (MVGs) $\mathbf{x}_1$ and $\mathbf{x}_2$ with $\mathbf{x}_1 \sim \mathcal{N}(\mu_{\mathbf{x}_1}, \Sigma_{\mathbf{x}_1})$ and $\mathbf{x}_2 \sim \mathcal{N}(\mu_{\mathbf{x}_2}, \Sigma_{\mathbf{x}_2})$, respectively, and assuming that $\Sigma_{\mathbf{x}_1}$ and $\Sigma_{\mathbf{x}_2}$ are non-singular, there exists a closed-form solution for the optimal transport operator, which is deterministic and linear:

$$\mathrm{T}^{\mathrm{MVG}}_{p_{\mathbf{x}_1} \longrightarrow p_{\mathbf{x}_2}}(x_1) = \Sigma_{\mathbf{x}_1}^{-\frac{1}{2}} \left( \Sigma_{\mathbf{x}_1}^{\frac{1}{2}} \Sigma_{\mathbf{x}_2} \Sigma_{\mathbf{x}_1}^{\frac{1}{2}} \right)^{\frac{1}{2}} \Sigma_{\mathbf{x}_1}^{-\frac{1}{2}} \cdot (x_1 - \mu_{\mathbf{x}_1}) + \mu_{\mathbf{x}_2}, \tag{2}$$

where a symmetric and positive definite square root of the matrices is chosen.

## 3.2 Wasserstein-2 MSE tradeoff

We build upon the problem setting introduced in [1, 2] to establish our analysis. We consider the following scenario: $\mathbf{x} \in \mathbb{R}^n$ represents a source natural image, $\mathbf{y} \in \mathbb{R}^m$ represents its degraded version, and we assume that the posterior $p_{\mathbf{x}|\mathbf{y}}(\cdot|y)$ is non-degenerate for almost any $y$. Our objective is to construct an estimator $\hat{\mathbf{x}}$ that predicts $\mathbf{x}$ given $\mathbf{y}$. A valid estimator $\hat{\mathbf{x}}$ should be independent of $\mathbf{x}$ given $\mathbf{y}$. Finally, $p_{\mathbf{x}}$, $p_{\mathbf{x}^*}$ and $p_{\hat{\mathbf{x}}_0}$ denote the probability distributions associated with the random variables $\mathbf{x}$, $\mathbf{x}^*$ and $\hat{\mathbf{x}}_0$, respectively.

Let $\mathbf{x}^* = \mathbb{E}[\mathbf{x}|\mathbf{y}]$ denote the MMSE estimator that achieves the minimal MSE, *i.e.*, $\text{MSE}(\mathbf{x}, \mathbf{x}^*) = D_{\min}$. Additionally, let $\hat{\mathbf{x}}_0$ denote the $\mathbf{D_{max}}$ estimator, which among all estimators satisfying $\mathcal{W}_2(p_{\mathbf{x}}, p_{\hat{\mathbf{x}}_0}) = 0$, attains the minimal MSE, namely, $\text{MSE}(\mathbf{x}, \hat{\mathbf{x}}_0) = D_{\max}$ (refer to fig. 1). Notably, as discussed in [1], these estimators have a compelling property: their joint distribution $p_{\hat{\mathbf{x}}_0, \mathbf{x}^*}$ is an optimal transport plan between $\mathbf{x}^*$ and $\mathbf{x}$, characterized by the following optimization problem:

$$p_{\hat{\mathbf{x}}_0, \mathbf{x}^*} \in \underset{p_{\mathbf{x}_1, \mathbf{x}_2} \in \mathbf{\Pi}(p_{\mathbf{x}}, p_{\mathbf{x}^*})}{\arg\min} \mathbb{E}\left[\|\mathbf{x}_1 - \mathbf{x}_2\|_2^2\right]. \tag{3}$$

In other words, finding $\hat{\mathbf{x}}_0$ is equivalent to finding an optimal transport plan from $\mathbf{x}^*$ to $\mathbf{x}$. Then, the $\mathbf{D_{max}}$ estimator is simply $\hat{\mathbf{x}}_0 = \mathrm{T}_{p_{\mathbf{x}^*} \longrightarrow p_{\mathbf{x}}}(\mathbf{x}^*)$.

This estimator is particularly useful as it allows to obtain any point on the perception-distortion function through a naive linear interpolation with the MMSE estimator. Specifically, we can define the interpolated estimator $\hat{\mathbf{x}}_P$ as follows:

$$\hat{\mathbf{x}}_P = (1 - \alpha)\hat{\mathbf{x}}_0 + \alpha\mathbf{x}^*, \tag{4}$$

where $0 \leq \alpha \leq 1$ is an interpolation constant [1] that depends on the perceptual index of $\mathbf{x}^*$ and the desired perceptual index $0 \leq P = \mathcal{W}_2(\mathbf{x}, \hat{\mathbf{x}}_P)$ (refer to fig. 1).

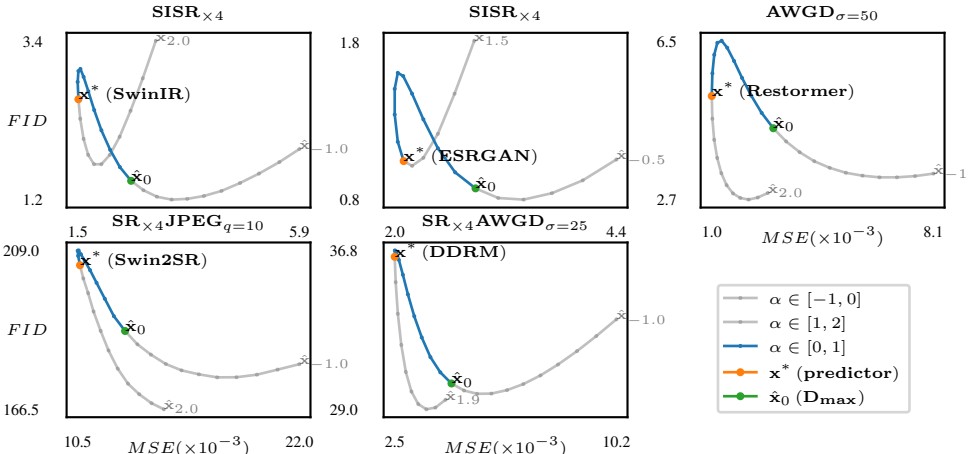

Figure 3: Trading perception and distortion using out-of-the-box predictors, wrapped with our method. Using eq. (4) with $\alpha \in [0, 1]$ we interpolate a given predictor (orange) and our improved $\mathbf{D_{max}}$ estimation (green), to approximate the PD FID-MSE function (blue curve). With $\alpha \in [-1, 0] \cup [1, 2]$ we extrapolate outside of the PD curve (light gray), beyond the theory-inspired area, to further improve performance.

## 4 Method

We start by describing the general flow of our proposed algorithm, and then move to elaborate on each of its components.

Theoretically speaking, our algorithm, combined with any given MMSE estimator, is an approximation of the $\mathbf{D_{max}}$ estimator. In practice, however, it can be combined with any type of estimator and potentially improve its perceptual quality. I.e., it can be combined with an estimator that optimizes

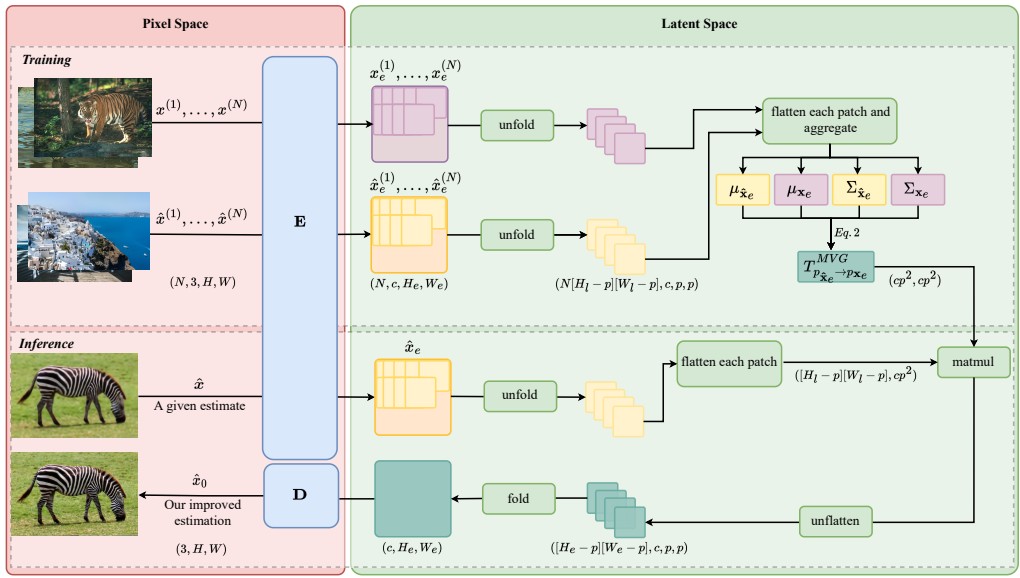

Figure 4: With a pre-trained VAE, we estimate the first and second order statistics of the latent patches of natural images and the restorations of some given estimator. At inference time, we use the closed-form OT eq. (2) operator between MVG distributions to transport the latent representation of a given restored sample, which, after decoding, increases the visual quality of the restored sample. For a fully detailed explanation of the algorithm, see section 4.

distortion (e.g., SwinIR [11]) and improve its perceptual quality at the expense of distortion, or it can be combined with an estimator that optimizes perceptual quality (e.g., DDRM [3]) and improve its perceptual quality even further. As a result, our algorithm is agnostic to the type of degradation. To clarify, our algorithm is not really a restoration algorithm by itself, but rather a wrapper which can potentially improve the perceptual quality of any given estimator.

## 4.1 The algorithm

The main goal of our algorithm is to approximate the optimal transport plan between $p_{\hat{\mathbf{x}}}$ and $p_{\mathbf{x}}$, namely, $T_{p_{\hat{\mathbf{x}}} \longrightarrow p_{\mathbf{x}}}$, where $\hat{\mathbf{x}}$ is a given estimator. Theoretically, with such an operator, one could optimally transport $\hat{\mathbf{x}}$ such that, with minimal loss in MSE performance, the transported estimator would attain perfect perceptual quality. Computing $T_{p_{\hat{\mathbf{x}}} \longrightarrow p_{\mathbf{x}}}$ for high dimensional distributions is a difficult task, involving complex (often adversarial) optimization (see discussion in section 2). To solve this, we perform several assumptions and approximations that allow us to efficiently compute a closed form transport operator that approximates $T_{p_{\hat{\mathbf{x}}} \longrightarrow p_{\mathbf{x}}}$. The flow of the algorithm is presented in fig. 4, and goes as follows:

**Encoding**: In the training stage we encode $N$ natural images $\{x^{(i)}\}_{i=1}^N$ and $N$ restored samples $\{\hat{x}^{(i)}\}_{i=1}^N$ (unpaired) into their latent representations, $\{x_e^{(i)}\}_{i=1}^N$ and $\{\hat{x}_e^{(i)}\}_{i=1}^N$, respectively. The size of each image sample is denoted by $(3, H, W)$, where $H$ and $W$ are the height and width, respectively, and the size of their latent representation is denoted by $(c, H_e, W_e)$. In the inference stage we perform the same process but only on a single estimate $\hat{x}$, resulting again in a latent representation of size $(c, H_e, W_e)$

**Unfold**: From each latent representation we extract all the overlapping patches of size $(c, p, p)$, where $p$ is the height and width of each patch.

**Flatten each patch and aggregate**: We flatten all the extracted patches to obtain 1-dimensional vectors of size $cp^2$, which we assume to come from a MVG distribution. In the training stage we compute their empirical mean and covariance matrix (aggregating over the $N$ dimension), and compute the optimal transport in closed-form $T_{p_{\hat{\mathbf{x}}_e} \longrightarrow p_{\mathbf{x}_e}}^{\text{MVG}}$ using eq. (2).

**Matmul and unflatten**: We apply the pre-computed transport operator using a simple matrix-vector multiplication on the flattened version of the patches extracted from the latent representation $\hat{x}_e$. We then reshape each vector to the original patch size $(c, p, p)$ (unflatten).

**Fold**: We rearrange the transported patches back to the original size of the latent representation (reversing the unfold operation). Since the patches overlap, we simply average the shared pixels.

**Decoding**: To produce our final enhanced estimation $\hat{x}_0$, we decode the transported latent image back to the pixel space using the decoder of the VAE.

Together, the inference steps form an end-to-end approximation of the desired transport operator $T_{p_{\hat{x}} \longrightarrow p_x}$. In appendix B we elaborate on the choices and practical considerations of our algorithm.

## 5 Experiments

In all of our experiments we use the encoder and decoder of the VAE [27] from stable-diffusion [7].

**The pre-trained models we evaluate:** We apply our latent transport method (described in section 4) on SwinIR [11], Restormer [12] and Swin2SR [13], all of which attempted to minimize average pixel distortion using a supervised regression loss on paired image samples. Additionally, we apply our algorithm on models that are trained to achieve high perceptual quality, and show that we can improve their visual quality even further. As such, we tested two benchmark models in high perceptual quality image restoration: ESRGAN [16], a GAN-based method, and DDRM [3], a diffusion-based method. Beyond our original goal to improve perceptual quality, we demonstrate that we can also traverse the $\mathcal{W}_2$ -MSE tradeoff using any restoration model (e.g., SwinIR, ESRGAN). To do so, we pick any of the aforementioned algorithms and apply our method to improve its perceptual quality, leading to a new estimator. We then interpolate the original algorithm and its improved version using eq. (4), adjusting $\alpha$ to traverse the tradeoff. To clarify, we plug the original algorithm as $\mathbf{x}^*$ in eq. (4) (instead of the theoretical MMSE estimator), and plug our improved version as $\hat{\mathbf{x}}_0$ (instead of the theoretical $\mathbf{D_{max}}$ estimator).

**Restoration tasks:** We showcase our algorithm on Single-Image Super-Resolution (SISR), denoising of Additive White Gaussian Noise (AWGN), JPEG [28] decompression, Noisy Super-Resolution (NSR) and Compressed Super-Resolution (CSR). Training and inference of our algorithm are performed on each restoration model separately, and the evaluation is performed on the restoration task that corresponds to the given model.

**Transport operator computation:** The transport operator is computed using two disjoint sets of 10 randomly picked images from the ImageNet [29] dataset train split. The first set is used to approximate the predictor's latent statistics $(\mu_{\hat{\mathbf{x}}_e}, \Sigma_{\hat{\mathbf{x}}_e})$: we degrade each image according to the restoration task the predictor is intended to solve, compute the 10 restored outputs, embed the results and compute the embeddings' statistics. The second set is embedded into the latent representation without further modification and serves to approximate the natural image latent statistics $(\mu_{\mathbf{x}_e}, \Sigma_{\mathbf{x}_e})$.

**Metrics:** In addition to Peak Signal-to-Noise Ratio (PSNR), we evaluate distortion performance with Structural Similarity Index Measure (SSIM) [30] and Learned Perceptual Image Patch Similarity (LPIPS) [31], both of which suit better for natural image comparison [31].

Nonetheless, LPIPS remains by definition a *distortion* (full-reference) measure: it is non-negative and zero when the two images are identical [2]. Interestingly, the original perception-distortion paper [2] already classified the VGG loss [32] - the ancestor of LPIPS - to be a distortion, on which the tradeoff exists (but is less severe).

Therefore, to evaluate perceptual quality, we use the Inception Score (IS) [33], the Fréchet Inception Distance (FID) [25] and the Kernel Inception Distance (KID) [34] following popular image restoration papers [3, 7, 35].

**Data sets:** It is impractical to perform a serious quantitative evaluation of the perception-distortion tradeoff on real-world datasets (e.g., SIDD, DND, RealSR), which have too few samples to compute FID. Hence, for all models except DDRM [3] and Swin2SR [13], we report the performance on the 50,000 validation samples of ImageNet [29] following [7, 35]. Because of its computational complexity, DDRM [3] reported its performance on a subset of a 1000 ImageNet [29] validation samples. For Swin2SR [13], we use the official DIV2K [36] restored samples provided by the authors.

Although our algorithm can be applied to images with arbitrary aspect ratios, all the tested models were trained on square images. Thus, we resize the samples to $512 \times 512$ pixels following the pre-processing procedure of DDRM [3]. Finally, we conduct the qualitative evaluation on popular samples from DIV2K or Set14.

## 5.1 Quantitative results

As reported in table 1, our algorithm can trade perceptual quality for distortion (and vice versa) at test time. We sometimes even manage to improve the pre-trained predictor's PSNR, even of regression models like SwinIR and Swin2SR. When using $\alpha \leq 0$, we systematically improve the predictor's perceptual performance (FID, KID, IS), even for estimators which were designed to achieve photo-realism in the first place, e.g., ESRGAN and DDRM. On Non-Local-Means (NLM) [15], an older, non deep-learning denoising algorithm, our method marginally improves all metrics.

While, in theory, our procedure to traverse the perception distortion tradeoff should only include values of $\alpha$ in the range $[0, 1]$ (see eq. (4)), we also tried to use values outside of this range. As shown in fig. 3, with $\alpha \in [-1, 0] \cup [1, 2]$ we can obtain even better PD curves, and sometimes improve the perceptual quality and/or the distortion of the methods even further. For instance, the PD curve of Swin2SR obtained using $\alpha \in [1, 2]$ is strictly better than the one obtained using $\alpha \in [0, 1]$. This deviation from the theory can be explained by the implementation choices discussed in section 4; We perform the transport in the latent space – not the pixel space. Additionally, we use FID as perceptual index to measure visual quality when the theory presented in section 3.2 only talks about the Wasserstein-2 distance. In practice, the sharpened details that appear in $\hat{\mathbf{x}}_0$ and not in $\mathbf{x}^*$ are either amplified when $\alpha \in [-1, 0]$ or subtracted (instead of being added) to $\mathbf{x}^*$ when $\alpha \in [1, 2]$. We leave the formal analysis of this interesting phenomenon for future research.

**Choosing the right value of** $\alpha$: Like any other hyper-parameter, $\alpha$ can improve the performance with some tuning when approaching a new task or a new data set (refer to table 1). We argue that the few-shot nature of our algorithm makes this tuning actually practical: $\alpha$ does not need to be set before performing some expensive training. Once $\hat{\mathbf{x}}_{\alpha=0}$ is computed, any $\hat{\mathbf{x}}_\alpha$ can be obtained thanks to eq. (4) without additional cost. In any case, as reported in table 1, $\alpha = 0$ consistently improves perceptual quality for all the tasks and models considered (as expected from the theory). We consider it to be a satisfying default choice, so manually adjusting $\alpha$ is not a great concern.

## 5.2 Qualitative results

Qualitative results on arbitrary image sizes and aspect ratios are shown in fig. 5. Using our method, we observe a consistent improvement of photo-realism when transporting existing restoration algorithms using our method. Hence, the qualitative results align with the quantitative perceptual performance gains.

## 5.3 Training details & ablation study

All the results presented in figs. 3 and 5 and table 1 were obtained using the same hyper-parameters. We used the "f8-ft-MSE" fine-tuned version of stable-diffusion's VAE from Hugging-Face's diffusers library [37]. For the training stage we use 20 randomly-drawn images from the ImageNet training set (10 images which we use as the natural images set, and 10 images which we degrade and then restore with the estimator). We used a patch-size of $p = 3$ in the latent space.

Thanks to its simplicity, for each restoration task, our few-shot algorithm requires just a single GPU, and a few seconds for both training and inference.

We turn to detail some considerations about practical aspects of our algorithm which we empirically evaluate on the popular SISR$_{\times 4}$ task for the ESRGAN estimator.

**Patch size**: We experiment with increasing patch-sizes when unfolding the latent image (see appendix B.2). $p = \{3, 5\}$ yielded the best PSNR and FID. Smaller patch size ($p = 1$) resulted in worse FID and bigger size $7 \leq p \leq 15$ yielded slightly worse PSNR.

**Training size**: As discussed in appendix B.4, each image contributes thousands of samples to the computation of the OT operator. Still, we expect the empirical statistics estimation to benefit from a larger sample size $S$. To confirm this, we repeated the visual enhancement experiments while

Table 1: Using eq. (4), our algorithm can trade-off perception and distortion at inference time on any predictor [3, 11–13, 15, 16] and image restoration task. For each task, we report the performance of $\hat{\mathbf{x}}_0$. It consistently improves perceptual metrics on all taks and models (aside of NLM). We also report other interesting choices of $\alpha$ that optimize perception and distortion (for more details about this choice refer to section 5.1).

| Task | Signal | Distortion | | | Perception | | |
|---|---|---|---|---|---|---|---|
| | | PSNR ↑ | SSIM ↑ | LPIPS ↓ | FID ↓ | IS ↑ | KID×$10^3$ ↓ |
| | $\mathbf{x}$ | $\infty$ | 1 | 0 | 0 | 240.53±4.42 | 0 |
| | $\mathbf{D}(\mathbf{E}(\mathbf{x}))$ | 27.10 | 0.81 | 0.13 | 0.24 | 234.71±4.04 | 0.02±0.07 |
| SISR$_{\times 4}$ | SwinIR [11] | 28.10 | **0.84** | **0.24** | 2.54 | 201.52±4.85 | 1.24±0.24 |
| | $\hat{\mathbf{x}}_{0.9}$ | **28.15** | **0.84** | **0.24** | 2.80 | 198.69±2.97 | 1.38±0.24 |
| | $\hat{\mathbf{x}}_{-0.2}$ | 25.08 | 0.77 | 0.25 | **1.19** | **216.74**±4.26 | **0.38**±0.89 |
| | $\hat{\mathbf{x}}_0$ | 25.48 | 0.78 | 0.23 | 1.39 | 214.63±5.50 | 0.69±0.23 |
| JPEG$_{q=10}$ | SwinIR [11] | **29.68** | **0.86** | **0.30** | 8.95 | 161.73±3.36 | 6.52±0.77 |
| | $\hat{\mathbf{x}}_{1.1}$ | 29.58 | **0.86** | **0.30** | 8.36 | 166.50±3.12 | 6.08±0.75 |
| | $\hat{\mathbf{x}}_{-0.2}$ | 23.74 | 0.76 | 0.31 | **7.56** | **166.65**±3.58 | **5.68**±0.83 |
| | $\hat{\mathbf{x}}_0$ | 24.84 | 0.78 | 0.30 | **8.14** | **163.14**±3.93 | **6.15**±0.77 |
| AWGN$_{\sigma=50}$ | Restormer [12] | **30.18** | **0.86** | 0.26 | 5.21 | 178.62±2.83 | 3.29±0.56 |
| | $\hat{\mathbf{x}}_{1.1}$ | 30.09 | **0.86** | **0.25** | 4.63 | 183.36±3.20 | 2.61±1.53 |
| | $\hat{\mathbf{x}}_{1.7}$ | 27.26 | 0.82 | **0.25** | **2.73** | **198.93**±5.13 | **1.76**±1.58 |
| | $\hat{\mathbf{x}}_0$ | 25.31 | 0.78 | 0.27 | 4.42 | **182.86**±2.21 | 2.93±1.62 |
| SR$_{\times 4}$JPEG$_{q=10}$ | Swin2SR [13] | 19.75 | **0.55** | 0.53 | 205.00 | 5.95±0.49 | 40.68±3.34 |
| | $\hat{\mathbf{x}}_{0.8}$ | **19.81** | **0.55** | 0.53 | 209.82 | 5.91±0.69 | 43.28±3.86 |
| | $\hat{\mathbf{x}}_{1.9}$ | 18.44 | 0.49 | **0.51** | **168.12** | 6.36±0.69 | **19.95**±2.84 |
| | $\hat{\mathbf{x}}_0$ | 18.45 | 0.48 | **0.51** | 183.80 | **6.55**±0.61 | 29.07±3.58 |
| SISR$_{\times 4}$ | ESRGAN [16] | 26.77 | 0.80 | **0.21** | 1.06 | 221.68±3.06 | 0.43±0.14 |
| | $\hat{\mathbf{x}}_{0.7}$ | **27.00** | **0.81** | **0.21** | 1.51 | 215.87±3.64 | 0.56±0.21 |
| | $\hat{\mathbf{x}}_{-0.2}$ | 24.84 | 0.74 | 0.23 | **0.80** | **221.89**±2.53 | **0.30**±0.20 |
| | $\hat{\mathbf{x}}_0$ | 25.33 | 0.74 | 0.22 | 0.89 | 220.96±3.19 | 0.34±0.18 |
| SR$_{\times 4}$AWGN$_{\sigma=50}$ | DDRM [3] | **26.10** | **0.75** | 0.34 | 36.44 | 43.52±3.33 | 5.09 |
| | $\hat{\mathbf{x}}_{1.2}$ | 25.91 | **0.75** | **0.33** | 33.68 | 44.90±4.06 | 3.88 |
| | $\hat{\mathbf{x}}_{1.7}$ | 24.48 | 0.70 | 0.35 | **29.05** | **47.91**±2.69 | **1.47** |
| | $\hat{\mathbf{x}}_0$ | 23.19 | 0.69 | 0.35 | 29.71 | 46.36±4.18 | 1.91 |
| AWGN$_{\sigma=50}$ | NLM [15] | 26.09 | 0.71 | 0.44 | 12.84 | 148.71±3.75 | 8.73±0.91 |
| | $\hat{\mathbf{x}}_{0.8}$ | **26.24** | **0.72** | **0.43** | **12.46** | **148.78**±2.49 | **8.60**±0.95 |
| | $\hat{\mathbf{x}}_0$ | 24.80 | 0.71 | **0.42** | 14.88 | 140.65±2.10 | 10.90±1.15 |

varying the number of training samples. Surprisingly, we observe no change in the performance of the evaluated metrics for $S = \{10^5, 10^4, 10^3, 10^2\}$, i.e., approximating the distribution statistics with 100 samples is as good as using 100,000 samples, and this is true regardless of the chosen patch size. Moreover, when With $S = 10$, then only for small patch sizes of $p \leq 5$ we observe no performance drop compared to using a larger sample size. This suggests that our method can be successfully deployed in few-shot settings, where the number of available samples is small.

**Paired vs. unpaired samples**: Surprisingly, using paired images to compute the distribution parameters yielded better PSNR but worse FID. We suspect that using paired updates induces a bias which results in worse covariance estimation.

**Transporting the degraded measurement directly**: Applying our algorithm on the degraded input directly led to insufficient results as we see in appendix B.7.

**Re-applying the algorithm another time on $\hat{\mathbf{x}}_0$**: This is actually an interesting idea we tested on super-resolution when conducting our evaluations. As a matter of fact, the performance does not improve (it even degrades a bit) when applying the algorithm another time. The explanation is quite simple: After transporting once the test images using the VAE, their latent distribution aligns with that of the natural images. Hence, transporting another time does nothing (the transport operator is

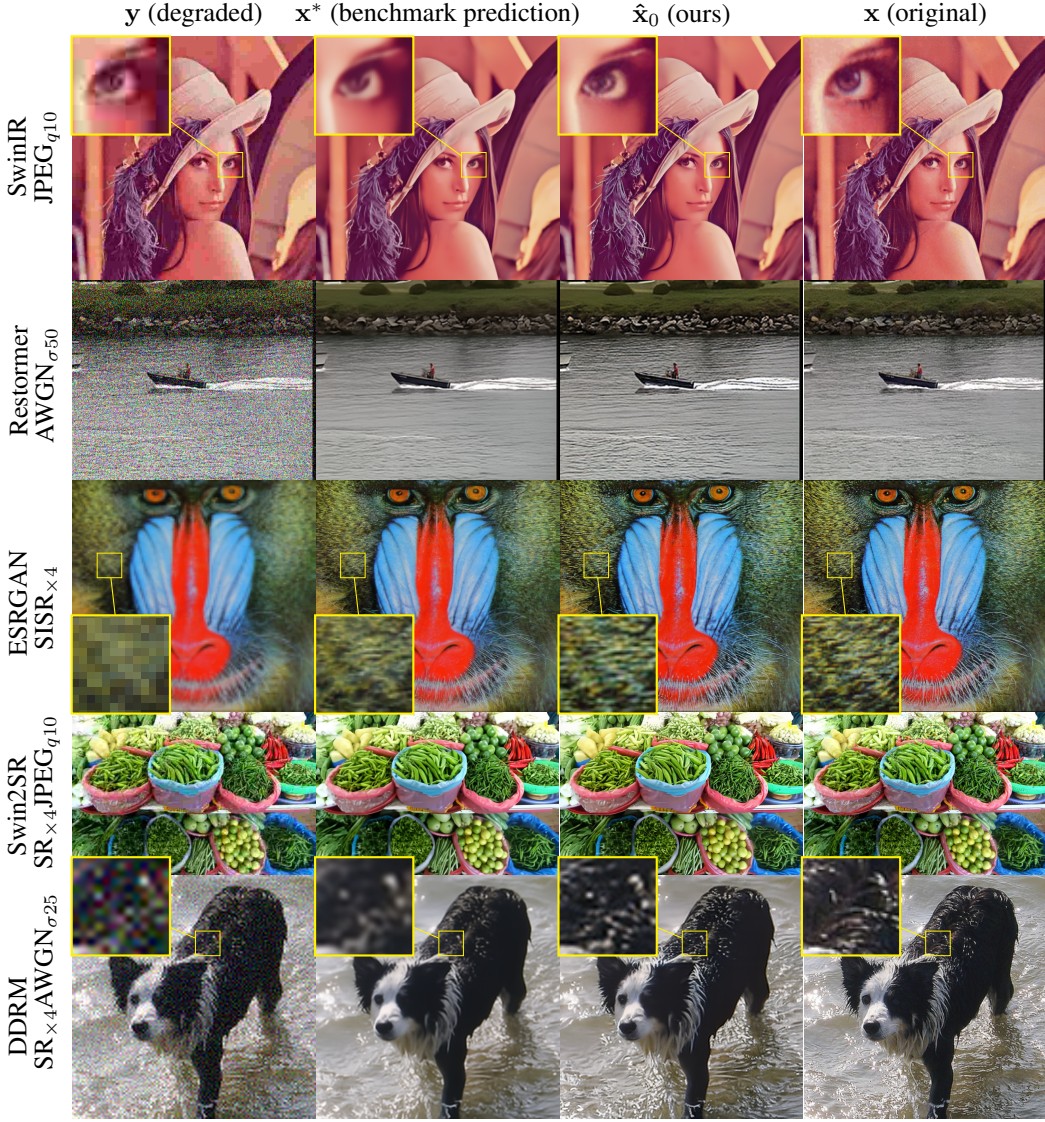

Figure 5: Our method (third column from the left) notably improves the results of several benchmark predictors (second column from the left) on various degradations.

the identity matrix). We are only left with the reconstruction error introduced by the encoding and decoding of the images, which deteriorates the MSE performance.

**Does the selection on the training data have an impact on the performance of restoration?**: Our experiments showed that the class of images does not have a significant impact on the performance (e.g. one could use images of cars to improve images of dogs). However the resolution of images does play a significant role in attaining the best performance. I.e., to transport 512x512 images, it is best to use training images of the same resolution. This drawback is somewhat mitigated by the few-shot nature of the algorithm.

## 6 Discussion

**Limitations**: The pre-trained VAE used for the purpose of our experiments exhibits a rate of $R = 48$ on $512^2$ images which inevitably translates into sub-optimal distortion performance [38]. Thus, the distortion performance of our estimates are bounded by that of the pre-trained VAE. I.e., even encoding and decoding a completely clean and natural image does not yield result in perfect

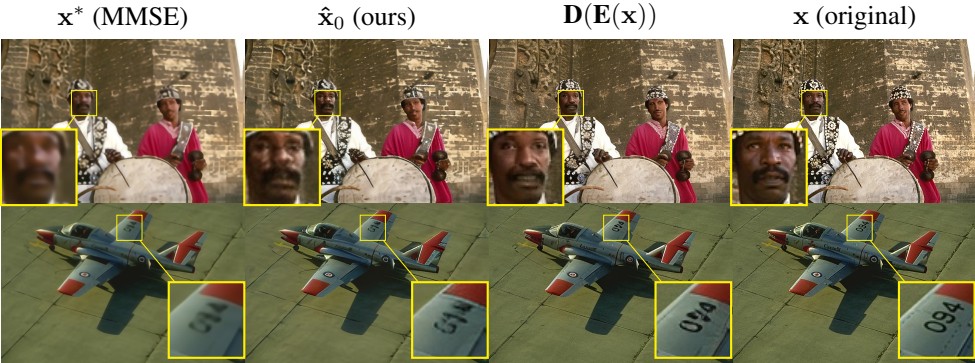

| $\mathbf{x}^*$ (MMSE) | $\hat{\mathbf{x}}_0$ (ours) | $\mathbf{D}(\mathbf{E}(\mathbf{x}))$ | $\mathbf{x}$ (original) |

Figure 6: Our method's reconstruction capabilities are bounded by that of the VAE. Our algorithm is not able to preserve complex visual structures such as face identity (top row) or text (middle row).

reconstruction. Most notably, the VAE sometimes fails to reconstruct human faces, as well as text images, and such a weaknesses affects our algorithm as well (see fig. 6).

Finally, it has been recently shown that the posterior sampler is the only estimator attaining perfect perceptual quality while producing outputs that are perfectly consistent with the degraded input [39]. As such, $\mathbf{D}_{\mathbf{max}}$ cannot hope for consistent restorations.

**Potential impact**: Instead of using sophisticated and data-hungry generative models, we show it is possible to obtain photo-realistic results using simple tools like MMSE estimators and VAEs. We hope our few-shot algorithm will inspire other simple and practical image restoration methods.

**Potential misuse**: Our algorithm aims at improving the perceptual quality of existing algorithms. However, when using a biased training set, this could potentially cause bias in the enhanced restoration as well. This could potentially harm the results of medical image diagnosis, for example.

## Acknoledgements

This research was partially supported by the Council For Higher Education - Planning and Budgeting Committee.

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

# Deep Optimal Transport: A Practical Algorithm for Photo-realistic Image Restoration - Supplementary Material

## A  Background and extensions

### A.1  Numerical Example

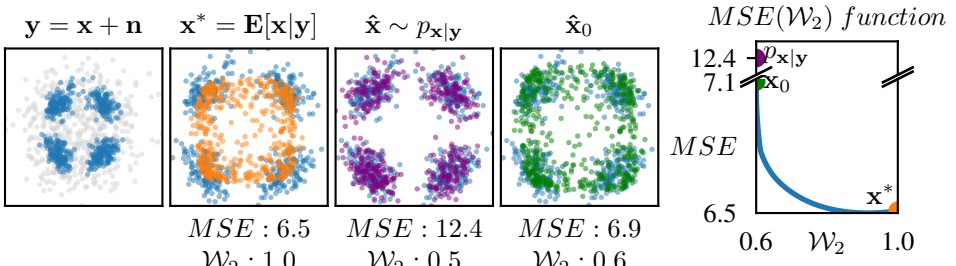

Figure 7:  2D Gaussian mixture denoising. Source samples are shown in blue. The $MMSE$ estimator ($\mathbf{x}^*$, orange) attains the best $MSE$ but the worst perceptual index $\mathcal{W}_2$. The posterior samples ($\mathbf{x}|\mathbf{y}$, purple) attain the best perceptual index but half of the optimal $MSE$ performance. The $\mathbf{D_{max}}$ estimator ($\hat{\mathbf{x}}_0$, green) maintains the $MSE$ of $\mathbf{x}^*$ while attaining a perceptual quality close to $\mathbf{x}|\mathbf{y}$. The DP curve is obtained by interpolating $\hat{\mathbf{x}}_0$ and $\mathbf{x}^*$ using eq. (4).

To guide the reader in understanding the MMSE transport paradigm, we showcase our method on a 2-dimensional denoising problem. To avoid a too trivial uni-modal example, we draw the clean signal from a 4-components Gaussian mixture with non-trivial covariances. We derive linear MMSE and posterior estimators from [40] and proceed by applying the closed-form transport operator introduced in eq. (3).

Note that to avoid deviating from our actual method, we refrain from using more advanced transport operators better suited for multi-modal data. Indeed, those are not a practical solution for real-world image datasets, as they require much more samples than actually available.

We summarize the experiment results in fig. 7. We observe that we obtain the best perceptual quality by sampling from the posterior distribution. However, we witness a significant decrease in $MSE$ performance as predicted by [2]. In contrast, the $\mathbf{D_{max}}$ estimator enjoys a good perceptual index while maintaining a close-to-optimal distortion performance.

### A.2  Stochastic transport operator

Throughout our experiments, we found out that increasing the patch-size $p$ can result in numerical instabilities. Recall that the linear transport operator presented in eq. (3) uses the inverse square root of the source covariance matrix $\Sigma_{\mathbf{x}_1}$. When $p$ is large, (typically $p \geq 7$), we obtain ill-conditioned covariance matrices. When the smallest singular value is still positive, we add a small stability constant to the matrix diagonal to ensure it is strictly positive definite. However, the numerical errors sometimes adds up to negative eigenvalues. [2] In this case, we clamp the negative eigenvalues to zero and use the stochastic (one-to-many) transport operator proposed by [1],

$$\mathrm{T}^{\text{stochastic}}_{p_{\mathbf{x}_1} \longrightarrow p_{\mathbf{x}_2}}(x_1) = \Sigma_{\mathbf{x}_2}^{\frac{1}{2}} \left( \Sigma_{\mathbf{x}_2}^{\frac{1}{2}} \Sigma_{\mathbf{x}_1} \Sigma_{\mathbf{x}_2}^{\frac{1}{2}} \right)^{\frac{1}{2}} \Sigma_{\mathbf{x}_2}^{-\frac{1}{2}} \Sigma_{\mathbf{x}_1}^{\dagger}(x_1 - \mu_{\mathbf{x}_1}) + \mu_{\mathbf{x}_2} + w, \tag{5}$$

when $\Sigma_{\mathbf{x}_1}^{\dagger}$ denotes the pseudo-inverse of $\Sigma_{\mathbf{x}_1}$ (after negative eigenvalues where clamped) and $w \sim \mathcal{N}(0, \Sigma_{\mathbf{x}_2}^{\frac{1}{2}}(I - \Sigma_{\mathbf{x}_2}^{\frac{1}{2}} \mathrm{T}^* \Sigma_{\mathbf{x}_1}^{\dagger} \mathrm{T}^* \Sigma_{\mathbf{x}_2}^{\frac{1}{2}})^{\frac{1}{2}} \Sigma_{\mathbf{x}_2}^{\frac{1}{2}})$, with $\mathrm{T}^* = \Sigma_{\mathbf{x}_2}^{-\frac{1}{2}} \left( \Sigma_{\mathbf{x}_2}^{\frac{1}{2}} \Sigma_{\mathbf{x}_1} \Sigma_{\mathbf{x}_2}^{\frac{1}{2}} \right)^{\frac{1}{2}} \Sigma_{\mathbf{x}_2}^{-\frac{1}{2}}$.

---

[2]We tried to avoid overflow when summing over the images by using 64 bit precision

# B Practical choices and considerations in our algorithm

## B.1 Working in latent space

We adopt the latent transport approach where the images are embedded into the latent space of a pre-trained auto-encoder. Let $\mathbf{E}(\cdot)$, $\mathbf{D}(\cdot)$ denote the encoder and decoder, respectively. Even if $\mathbf{D}(\mathbf{E}(t)) = t$, it is likely that $\mathbf{E}(\cdot)$ "deforms" the space, I.e., $\|\mathbf{E}(s) - \mathbf{E}(t)\| \neq \|s - t\|$, which means that the optimal transport plan in the latent space could be *different* than the plan we seek in the pixel space (the cost function in eq. (3) has changed). We can address this by modifying the latent cost function to account for the deformation via the following change of variables

$$\mathbb{E}\left[\|\hat{\mathbf{x}} - \mathbf{x}\|^2\right] = \mathbb{E}\left[\frac{\|\mathbf{E}(\hat{\mathbf{x}}) - \mathbf{E}(\mathbf{x})\|^2}{|\frac{\partial \mathbf{E}(\mathbf{x})}{\partial \mathbf{x}}| \cdot |\frac{\partial \mathbf{E}(\hat{\mathbf{x}})}{\partial \hat{\mathbf{x}}}|}\right], \tag{6}$$

where $|\frac{\partial \mathbf{E}(\mathbf{x})}{\partial \mathbf{x}}|$ is the determinant of the Jacobian matrix of $\mathbf{E}(\cdot)$ However it is not a practical solution since we lose access to the closed-form solution eq. (2). Note that the latent MSE approximation is usually desirable when dealing with natural images (e.g. to elaborate image quality measure [41], perceptual quality metrics [25]). It is also true in our case but it means we can no longer claim we obtain the $\mathbf{D_{max}}$ estimator.

With that, we argue that switching to a latent cost is actually a strength rather than a weakness of our method. Indeed, using the MSE between deep latent variables has shown to be a better fit to compare natural images than directly working in the pixel space [31]. The authors of [7] trained their VAE (which is used in our experiments) to remove "imperceptible details" from the latent representation, in order to better focus on higher level image semantics. In section 5.1 we validate this claim by showing that our algorithm maintains the "perceptual" discrepancy performance of the original estimator (*e.g.*, LPIPS).

## B.2 Overlapping patches extraction strategy

For Convolutional Neural Network (CNN) encoders [3], let $(c, H_e, W_e)$ denote the shape of the latent representation (CNN encoders produce 3-dimensional encoded tensors), where $H_e, W_e$ the spatial extent and $c$ is the number of channels (i.e., the number of convolution kernels in the last convolution layer). The covariance matrices $\Sigma_{\hat{\mathbf{x}}_e}$, $\Sigma_{\mathbf{x}_e}$ contain $\frac{(c, H_e, W_e)^2}{2}$ parameters, which may require a large amount of samples for large latent images with $H_e, W_e \gg 1$. To mitigate the quadratic dependency on $H_e \cdot W_e$, we assume that the latent pixels depend only on the pixels in their close neighborhood. In practice, we unfold the latent representation, extracting all overlapping patches of shape $(c, p, p)$. A similar approximation exists in the style-transfer literature [8, 9], where instead of patches, only the pixels are considered (i.e., this is a private case of our approach with $p = 1$). In section 5.3 we empirically show that increasing $p$ improves the perceptual quality at the expense of MSE performance, given that enough training samples are available.

## B.3 Shared distribution

When dealing with natural image scenes, it is beneficial to suppose that overlapping patches share common statistical attributes [41, 42]. In the case of a CNN encoded image, this approximation remains satisfying because we ultimately look at filter activations which are spatial-invariant with each latent patch having the same receptive field. Therefore, we assume that the overlapping patches are all samples from the same distribution. This approach dramatically reduces the number of estimated parameters, and also multiplies the number of samples at our disposal by $H_e \cdot W_e$, which alleviates the curse of dimensionality. We demonstrate these practical benefits in section 5.3. In practice, given $N$ images, we "flatten" all the extracted patches to vectors $\underline{v}_{cp^2 \times 1}$ which we stack into a sample matrix $\underline{\underline{X}}_{NH_eW_e \times cp^2}$. We then aggregate the samples to compute the MVG statistics: $\mu = X^T \mathbf{1}$, $\Sigma = \frac{NH_eW_e}{NH_eW_e-1}(X - \mu)(X - \mu)^T$. As $NH_eW_e$ may be very large, we perform all computations in double precision. When training, this process is done twice; once for the natural image samples, and once for the restored samples we wish to transport.

---

[3]This methodology can easily be extrapolated to other encoder architectures.

## B.4  Size of the latent representation

When increasing the capacity of models with a fixed encoding rate, deepening is preferable than widening. Indeed, increasing $c$ makes the covariance estimation dramatically harder while increasing $H_e, W_e$ enlarges the sample pool. Therefore, the VAE from [7] with $c = 4$ and $H_e, W_e \gg 1$ is a particularly good candidate for our method. For $p = 3$ for instance, the covariance matrix admits only $1296$ parameters while each $512^2$ image contributes $4096$ samples to its estimation. As we see next, this greatly contributes to reducing the number of training samples needed to estimate the covariance matrices and allows to compute the transport operator in a few-shot manner.

## B.5  Transport

In a single pass on a data set of natural images and a (possibly different) data set of restored samples, we compute $T^{\text{MVG}}_{p_{\hat{\mathbf{x}}_e} \longrightarrow p_{\mathbf{x}_e}}$ (see eq. (2)). Note that each latent distribution could sometimes be degenerate, especially for severe degradations. Fortunately, the classical MVG transport operator can be generalized to ill-posed settings where $\Sigma_{\hat{\mathbf{x}}}$ is a singular matrix (see appendix A.2).

## B.6  Decoding

Since the transported patches overlap, we "fold" them back into a latent image $\hat{\mathbf{x}}_{0,latent}$ by averaging. The latent image is then decoded back to the pixel space, i.e. $\hat{\mathbf{x}}_0 = \mathbf{D}(\hat{\mathbf{x}}_{0,latent})$. Since $\mathbf{E}(\cdot)$ is not invertible, the decoder $\mathbf{D}(\cdot)$ is used as a convenient approximation in the training domain of the auto-encoder. A corollary of this approximation is that the auto-encoder should in theory be trained on the image distribution we aim to transport, which weakens our claim to a fully blind algorithm.

All the steps described above are summarized in fig. 4.

## B.7  Transporting the degraded measurement

We tried applying our algorithm on the degraded measurement directly. Indeed we observe qualitatively and quantitatively that transporting the degraded measurement $\mathbf{y}$ amplifies the degradation (refer to fig. 8).

| $\mathbf{y}$ (degraded) | $\mathbf{x}^*$ (SwinIR) | $T_{p_{\mathbf{y}} \longrightarrow p_{\mathbf{x}}}(y)$ | $\hat{\mathbf{x}}_0$ | $\mathbf{x}$ (original) |

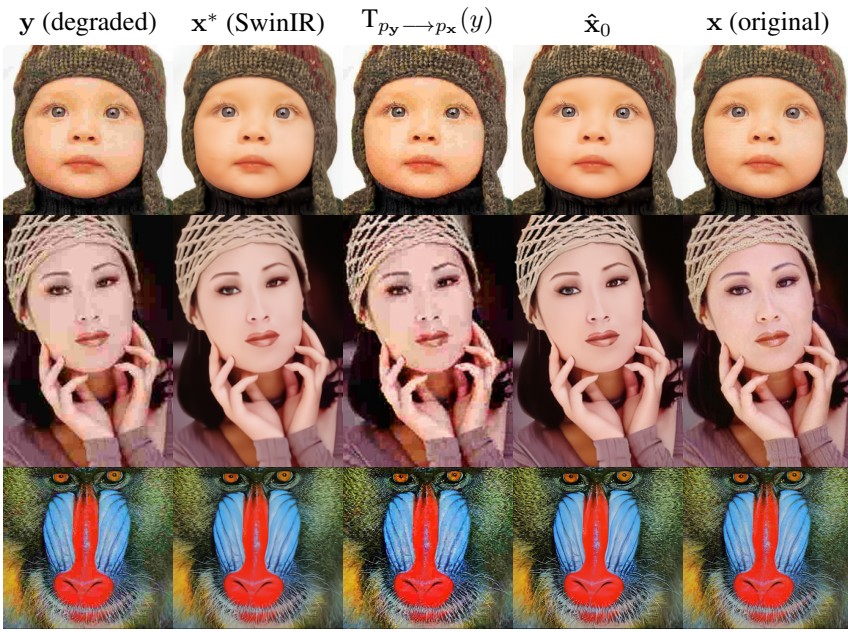

Figure 8: Transporting the degraded measurement ($\text{JPEG}_{q=10}$) directly is not enough to restore the image. It can sometimes even exacerbate the degradation. Quantitatively, the degraded sample $\mathbf{y}$ has better PSNR and FID than its transported version (respectively 27.26 dB and 13.88 FID *v.s.* 23.69 dB and 15.88 FID).

