# OpenReview forum: "Deep Optimal Transport: A Practical Algorithm for Photo-realistic Image Restoration"
_NeurIPS.cc/2023/Conference — NeurIPS 2023 poster_

### Official Review · Reviewer_XzV9 · 2023-07-06

**Soundness:** 2 fair
**Presentation:** 3 good
**Contribution:** 2 fair
**Rating:** 3
**Confidence:** 5

**Summary:**

This paper considers the high perceptual image restoration problem. The authors propose a method to control the tradeoff between the perceptual quality and distortion (e.g. MSE) of a pretrained restoration model. The mehtod is developed based on a recent theory [1] on the tradeoff between MSE distortion and perception quality measured by Wasserstein-2 distance, which indicates that minimum MSE restoration under perfect perception constraint can be achieved by an optimal transport from an minimum MSE estimator to a perceptual estimator. Meanwhile, in theory the perception-distortion tradeoff, with perception quality measured by Wasserstein-2 distance and distortion measured by MSE, can be controlled by a linear interpolation between an minimum MSE estimator and a perfect perceptual estimator.  Based on this theory, the authors propose to transport the output of a pretrained restoration model to improve the perceptual quality. The optimal transport is performed in the latent space under a Gaussian distribution assumption, with which closed-form optimal transport can be derived and, finally, the perception-distortion tradeoff is controlled by linearly transforming the first- and second-order statistics (means and covariances) in latent space of a pre-trained model. Experimental results on various image restoration tasks have been provided to demonstrate the effectiveness of the proposed method.


**Strengths:**

1. The proposed method is interesting, which can be viewed as a post-processing method that, given a pre-trained restoration model, it can achieve tradeoff between perception quality (measured by Wasserstein-2 distance) and distortion (measured by MSE) by a simple training stage computing empirical fisrt- and second-order statistics (means and covariances) in the latent space under a Gaussian distribution assumption.
2. This paper is clearly written and easy to read.
2. Experimental results verified the effectiveness of the proposed method.


**Weaknesses:**

1. Lack of theorectical novelty. The proposed method is based on the theory in [1] (see Section 3) and appears to be more of an extended evaluation of the results in [1]. Besides, the adopted appraoch that performs transport in the latent space is also not new and borrowed from existing work.

2. While the proposed method is interesting, its post-processing nature may limit its practical use since it requires a test-time training procedure. In comparison, one-stage restoration models would be preferred in practical applications. Although the test-time training only uses a dozen images restored by the pretrained model, retraining may be necessary for it to perform well when the degradation strength changes.

3. The assumption that the latent representations follow a Multivariate Gaussian distribution is relatively strong. The authors claim that their method can be applied to any pre-trained model. I’m afraid this assumption might limit its application as it requires the latent representation distribution to be (close to) Multivariate Gaussian.

4. Using optimal transport for image restoration is not new, e.g. for image denoising and super-resolution, which has demonstrated the capability of yielding high perception quality. A natural question is how does the proposed method compare with such one stage optimal transport based restoration approaches.




**Questions:**

The authors mentioned the limitation in restoring human faces and text images, of which the pixels are highly correlated with each other. Is it because the patch size set to $p={3,5}$ in this paper is relative small and fail to extract mutual information while transporting?

Another question is that the authors demonstrate that a larger patch size $7≤p≤15$ can yield slightly worse PSNR. Is it because the distribution is only close to MVG with small $p$? The assumption might be far from valid when $p$ gets larger, especially for face and text images, which are highly structured.


**Limitations:**

The authors have adequately addressed the limitations.

---

> ### Author Rebuttal · Authors · 2023-08-08
>
> > Lack of theorectical novelty. The proposed method is based on the theory in [1] (see Section 3) and appears to be more of an extended evaluation of the results in [1]. Besides, the adopted appraoch that performs transport in the latent space is also not new and borrowed from existing work.
>
> The reviewer is correct to point out that we extend the theory introduced by [(Freirich et. al)](https://proceedings.neurips.cc/paper/2021/hash/d77e68596c15c53c2a33ad143739902d-Abstract.html). Note however, that this work is only theoretical and does not propose a practical way to achieve the Dmax estimator.
> Similarly, we do not hide the fact that our latent transport approach is inspired by prior works (see lines 50-53).
> Nonetheless, we introduce several key-improvements to the naive channel-wise transport that enable our algorithm to improve perceptual quality of *any* restoration algorithm and *any* inverse problem task with few unpaired examples to train on. As far as we know, these achievements are unprecedented in the literature, and thus we are confident the novelty in our work is significant.
> We will make sure to emphasize this in the **related-work Section**.
>
> > While the proposed method is interesting, its post-processing nature may limit its practical use since it requires a test-time training procedure. In comparison, one-stage restoration models would be preferred in practical applications.
>
> The reviewer's standpoint is interesting because it contrasts with that of the other reviewers, which identified the plug-and-play property of our approach as a strength rather than a weakness. In the vast majority of applications, it seems sound to assume that we can obtain a dozen of unpaired images to apply the test-time procedure, in which case all the one-stage restoration models we considered were improved by our algorithm.
>
> > Although the test-time training only uses a dozen images restored by the pretrained model, retraining may be necessary for it to perform well when the degradation strength changes.
>
> As noted by the reviewer, we apply the algorithm at test-time, on the output of a given model. Interestingly, retraining is necessary only if the pre-trained model itself needs retraining, in which case the few-shot procedure is likely to consume a negligible fraction of the computation and time resources allocated to retrain the one-stage model. We will make sure to clarify this in the revised version of the paper.
>
> > The assumption that the latent representations follow a Multivariate Gaussian distribution is relatively strong. The authors claim that their method can be applied to any pre-trained model. I’m afraid this assumption might limit its application as it requires the latent representation distribution to be (close to) Multivariate Gaussian.
>
> Actually, we perform the transport in the latent space of a Variational Auto-Encoder, which is precisely trained to achieve a normally distributed latent representation. Additionally and as pointed out by the reviewer, Gaussian close-form transport in the latent space is not new, and has proven its stability over a wide range of application.
>
> > Using optimal transport for image restoration is not new, e.g. for image denoising and super-resolution, which has demonstrated the capability of yielding high perception quality. A natural question is how does the proposed method compare with such one stage optimal transport based restoration approaches.
>
> Since our algorithm is conceived to improve an already existing method, it cannot be compared toe-to-toe with other one-stage approaches (whether they are transport based or not). Nonetheless, we can apply our algorithm to improve further the performance of these methods.
>
> > The authors mentioned the limitation in restoring human faces and text images, of which the pixels are highly correlated with each other. Is it because the patch size set to $p=3,5$ in this paper is relative small and fail to extract mutual information while transporting?
>
> This is an interesting idea. Note however, that the Encoder used to embed the images admits a large receptive field (typically 64 pixels depending on the configuration). Hence, it is safe to assume that the spatial correlations are conserved even with a small latent patch-size. On the other hand, we show in the **Limitation Section** that human faces and text are already largely distorted by the Variational Auto-Encoder (VAE). I.e., encoding and then decoding a clean image containing text (without any transport) typically results in deteriorated text. Therefore, we deem more plausible that the problem resides in the VAE rather than the patch-size. This is an important clarification we will add to **Section 6**.
>
> > Another question is that the authors demonstrate that a larger patch size  can yield slightly worse PSNR. Is it because the distribution is only close to MVG with small $p$ ? The assumption might be far from valid when $7
> \leq p \leq 15$ gets larger, especially for face and text images, which are highly structured.
>
> As discussed earlier, we use the latent representation of a VAE, therefore latent images' prior follows a MVG. The reviewer is correct to point out that larger patch-size typically result in a deterioration in the approximated distribution. As explained in **Appendices B.3-B.4**, The dimensionality of the prior grows quadratically with the patch-size, and therefore the number of covariance matrix parameters is proportional to $p^4$. With that, the number of samples available to compute these parameters shrinks quadratically with $p$. In this regard, it is much more challenging to accurately approximate the transport operator for $7
> \leq p \leq 15$ compared to $p=3,5$.

---

> > ### Comment · Reviewer_XzV9 · 2023-08-20
> >
> > I thank the authors for their replies. I would like to keep my rating.

---

### Official Review · Reviewer_Md85 · 2023-07-06

**Soundness:** 2 fair
**Presentation:** 3 good
**Contribution:** 2 fair
**Rating:** 5
**Confidence:** 5

**Summary:**

This paper presents an image restoration algorithm targeting at further restore the processed images that have been restored by pre-trained restoration models. As most of the restoration tasks use MSE as the main criteria for restoration, the restored images tend to be blurred to achieve better PSNR performance. This work takes advantage of optimal transport from source to target distribution to improve the perceptual quality, where the optimal transport operator is computed from distribution mapping from the restored image to original images.

**Strengths:**

+ This work proposes an image enhancement method after the degraded images restored by pre-trained restoration models.
+ The tradeoff between MSE and perceptual quality is discussed and the balancing process between two criteria are formulated and visualized on multiple tasks.
+ Only a few unpaired images are required for the optimal transportation computing.


**Weaknesses:**

- As the proposed method uses a few images for optimal transport computing. How to guarantee that the distribution of test data is aligned with the training data. Does the selection on the training data have an impact on the performance of restoration?
- In table 1, the numerical results of restoration with different alpha settings are given. For different restoration tasks, the alpha is set to different values while comparison. What criteria are used to select the parameter?
- The experiments are conducted on ImageNet dataset. But for most restoration tasks, the restored images have relatively large resolution. Images in ImageNet dataset have considerable small resolution and its original images have been compressed with lossy compression. Why authors evaluate the method upon the task-specific datasets.
- The method has similar mechanism with flow-based image deblurring or image sharpening. The methods itself has limited restoration capabilities on degraded image restoration.


**Questions:**

What is the meaning of the term “D(E(x))” in table1 and figure6? Does it refer to the decoding results obtained from the encoded latent representation of x?

**Limitations:**

Authors have addressed the limitations of the method in the paper.

---

> ### Author Rebuttal · Authors · 2023-08-06
>
> > As the proposed method uses a few images for optimal transport computing. How to guarantee that the distribution of test data is aligned with the training data. Does the selection on the training data have an impact on the performance of restoration?
>
> Our experiments showed that the **class of images** does not have a significant impact on the performance (e.g. one could use images of cars to improve images of dogs). However the **resolution** of images does play a *significant* role in attaining the best performance. I.e., to transport `512x512` images, it is best to use training images of the same resolution. This drawback is somewhat mitigated by the few-shot nature of the algorithm. We thank the reviewer for drawing our attention to this topic, and we will make sure to include this discussion in the revised version of the paper.
>
> >In table 1, the numerical results of restoration with different alpha settings are given. For different restoration tasks, the alpha is set to different values while comparison. What criteria are used to select the parameter?
>
> We selected $\alpha$ by observing the interpolation & extrapolation curves in **Figure 3** and chose the values with the most significant effect on performance. Like any other hyper-parameter, $\alpha$ can improve performance with some tuning when approaching a new task or dataset. We argue that the few-shot nature of our algorithm makes this tuning actually practical (it does not need to be set before performing some expensive training). In any case, $\alpha=0$ consistently improves perceptual quality for all tasks and models considered (as expected from the theory). We consider it to be a satisfying default choice, such that manually adjusting $\alpha$ is not too great of a concern. As other reviewers pointed out the lack of consistency in selecting the reported values of $\alpha$, we will make sure to clarify this in the experiment section, and additionally report the performance for $\alpha=0$ in **Table 1**.
>
> > The experiments are conducted on ImageNet dataset. But for most restoration tasks, the restored images have relatively large resolution. Images in ImageNet dataset have considerable small resolution and its original images have been compressed with lossy compression. Why authors evaluate the method upon the task-specific datasets.
>
> As pointed out by another reviewer, the perception metrics FID, IS and KID are not stable on common restoration datasets with small-to-medium size. Hence, we perform our evaluation on the 50,000 Imagenet validation samples, following very popular image restoration papers like ([Saharia et. al, 2021](https://arxiv.org/abs/2104.07636),[Rombach et. al, 2021](https://arxiv.org/abs/2112.10752)). This is also why it is impractical to perform a serious quantitative evaluation of the perception-distortion tradeoff on real-world datasets (e.g., SIDD, DND, RealSR), which have too few samples. Finally, we conduct the qualitative evaluation on popular samples from DIV2K or Set14. This is actually not a trivial question that was raised by another reviewer. We shall include these clarifications after line `194` in the paper.
>
> > The method has similar mechanism with flow-based image deblurring or image sharpening. The methods itself has limited restoration capabilities on degraded image restoration.
>
> There is actually much similarity with even older, classical works which apply a carefully chosen linear transformation on all overlapping patches of the degraded image. We consider this as an advantage of our method rather than a drawback. Applying linear transformations in the latent space could prove to be a powerful yet simple approach with interesting properties like few-shot learning and robustness.
> Our algorithm is not designed to directly restore degraded images but rather improve the perceptual quality of **any** restoration model at test time. In this regard, we demonstrate significant performance gains on a wide range of models (regression, GANs, diffusion) and tasks (super-resolution, denoising, JPEG artifact removal etc.).

---

### Official Review · Reviewer_HWzm · 2023-07-07

**Soundness:** 3 good
**Presentation:** 3 good
**Contribution:** 3 good
**Rating:** 6
**Confidence:** 5

**Summary:**

This paper presents an image restoration algorithm that builds upon a trained network to further minimize MSE.
To achieve this goal, an optimal transport was approximated by a linear transformation in the latent space. Visual results show clear improvement by using the proposed approach.

**Strengths:**

- The idea is straightforward as optimizing delta is common in traditional image restoration
- The visual results are convincing

**Weaknesses:**

- The uncertain of the proposed idea is low so relatively the novelty is less.

**Questions:**

By applying the proposed algorithm another time, will the results be further improved?

**Limitations:**

None.

---

> ### Author Rebuttal · Authors · 2023-08-06
>
> > `[Summary]` This paper presents an image restoration algorithm that builds upon a trained network to further minimize MSE
>
> We would like to clarify that the main goal of our algorithm is actually to improve perceptual quality. As a side-effect, we discovered empirically that we could extend the theory introduced in [(Freirich et. al)](https://proceedings.neurips.cc/paper/2021/hash/d77e68596c15c53c2a33ad143739902d-Abstract.html) to also improve MSE, but this is not the main contribution of our paper.
>
> > The uncertain of the proposed idea is low so relatively the novelty is less.
>
> The main contribution of this work is a proposed algorithm that improves the perceptual quality of *any* restoration algorithm and *any* inverse problem task with few unpaired examples to train on. As far as we know, these achievements are unprecedented in the literature, and thus we believe that the novelty in this work is significant.
>
> > By applying the proposed algorithm another time, will the results be further improved?
>
> This is actually an interesting idea we tested on super-resolution when conducting our evaluations. As a matter of fact, the performance *does not* improve (it even degrades a bit) when applying the algorithm another time. The explanation is quite simple: After transporting once the test images using the VAE, their latent distribution aligns with that of the natural images. Hence, transporting another time does nothing (the transport operator is the Identity matrix). We are only left with the reconstruction error introduced by the encoding and decoding of the images, which degrades the MSE performance. We agree with the reviewer that this experiment adds to the reader's understanding and helps draw the limitations of our algorithm. We shall add it in the supplementary material.

---

> > ### Comment · Reviewer_HWzm · 2023-08-20
> >
> > Author's respond explains my question well. I'll stick to my original rating.

---

### Official Review · Reviewer_YhDf · 2023-07-07

**Soundness:** 3 good
**Presentation:** 3 good
**Contribution:** 2 fair
**Rating:** 5
**Confidence:** 4

**Summary:**

The paper proposes an image restoration algorithm that can control the perceptual quality and/or the mean square error (MSE) of any pre-trained model, trading one over the other at test time.

**Strengths:**

1. The method is plug-and-play, requires only a few samples, and does not require further training.
2. The method approximates the optimal transport by a linear transformation in the latent space of a variational auto-encoder, which is somewhat novel.

**Weaknesses:**

1. The main concern is in the experiments section.
(1) It would be better to evaluate the method on real-world datasets (e.g., SIDD, DND, and RealSR datasets), which may make more sense.
(2) Quantitative results of $\hat{x}_{0}$ should be given in Table 1.
(3) Quantitative results of some important ablation experiments (e.g., paired vs. unpaired samples, and transporting the degraded measurement directly) can be given.
(4) LPIPS is generally regarded as a perception metric in image restoration tasks. Since FID, IS and KID are not very stable, they are generally not used in image restoration tasks. Just looking at PSNR, SSIM and LPIPS, the method doesn't seem to achieve a good distortion-perception tradeoff.
2. The interpolation constant $a$ seems to be adjusted manually, which may be inflexible.
3. Projected distribution loss [1] and sliced Wasserstein loss  [2]  also show better distortion-perception tradeoff in image restoration tasks, although it needs to be used directly for training. I don't know if the authors have tried this way. Further elaboration may be needed on how the proposed method relates to and differs from this loss.

[1] Projected Distribution Loss for Image Enhancement. ICCP 2021.
[2] Self-supervised learning for real-world super-resolution from dual zoomed observations. ECCV 2022.

**Questions:**

Please see the weaknesses. I am willing to improve the score if the concerns are addressed well.

**Limitations:**

The limitations have been described in the paper.

---

> ### Author Rebuttal · Authors · 2023-08-06
>
> > (4) LPIPS is generally regarded as a perception metric in image restoration tasks. Since FID, IS and KID are not very stable, they are generally not used in image restoration tasks. Just looking at PSNR, SSIM and LPIPS, the method doesn't seem to achieve a good distortion-perception tradeoff.
>
> Please note that, *by definition*, LPIPS is a distortion metric, as it is evaluated on pairs of images. Interestingly, the original perception-distortion paper [(Blau & Michaeli, 2018)](https://openaccess.thecvf.com/content_cvpr_2018/html/Blau_The_Perception-Distortion_Tradeoff_CVPR_2018_paper.html) already classified the VGG loss - the ancestor of LPIPS - to be a distortion, on which the tradeoff exists. Also, the reviewer is correct to point out that FID, IS and KID are not stable on common restoration datasets, **but this is largely caused by these datasets' small size**.
>
> > (1) It would be better to evaluate the method on real-world datasets (e.g., SIDD, DND, and RealSR datasets), which may make more sense.
>
> Following the comment above, it is why we perform our evaluation on the 50,000 Imagenet validation samples, following very popular image restoration papers like ([Saharia et. al, 2021](https://arxiv.org/abs/2104.07636),[Rombach et. al, 2021](https://arxiv.org/abs/2112.10752)). This is also why it is impractical to perform a serious quantitative evaluation of the perception-distortion tradeoff on real-world datasets (e.g., SIDD, DND, RealSR), which have too few samples. LPIPS is a distortion, not a perceptual quality metric.
> Finally, we do conduct the qualitative evaluation on popular samples from DIV2K or Set14. This is actually not a trivial question that was raised by another reviewer. We shall include these clarifications after line 194.
>
> > (2) Quantitative results of $\hat{x}_0$ should be given in Table 1. (3) Quantitative results of some important ablation experiments (e.g., paired vs. unpaired samples, and transporting the degraded measurement directly) can be given.
>
> In **Table 1**, we preferred not overloading the reader with yet additional rows but rather have them focus on the way we can trade perception over distortion with different values of $\alpha$, which constitute the main result of the paper. However, concerns around the reported values of $\alpha$ were raised by other reviewers and we understand the importance of these results. Note that quantitative results of $\hat{x}_0$ are visible in **Figure 3**, but we shall add the exact performance in **Table 1**. We will also add the ablation figures in a complementary table in the appendix.
>
> > The interpolation constant $\alpha$ seems to be adjusted manually, which may be inflexible.
>
> Like any other hyper-parameter, $\alpha$ can improve performance with some tuning when approaching a new task or dataset. We argue that the few-shot nature of our algorithm makes this tuning actually practical (it does not need to be set before performing some expensive training). In any case, $\alpha=0$ consistently improves perceptual quality for all tasks and models considered (as expected from the theory). We consider it to be a satisfying default choice, such that manually adjusting $\alpha$ is not too great of a concern. As other reviewers pointed out the lack of consistency in selecting the reported values of $\alpha$, we will make sure to clarify this in the experiment section, and additionally report the performance for $\alpha=0$ in **Table 1**.
>
> > Projected distribution loss [1] and sliced Wasserstein loss [2] also show better distortion-perception tradeoff in image restoration tasks, although it needs to be used directly for training. I don't know if the authors have tried this way. Further elaboration may be needed on how the proposed method relates to and differs from this loss.
>
> The paper is interested in the MSE-W2 tradeoff following the transport theorem introduced by [(Freirich et. al, 2021)](https://proceedings.neurips.cc/paper/2021/hash/d77e68596c15c53c2a33ad143739902d-Abstract.html). From a theoretical standpoint, it is not clear at all how we can obtain the Dmax estimator $\hat{x}_0$ using projected losses. In practice, we carefully designed our algorithm to improve existing restoration models at test time. Therefore, the aforementioned losses are not obvious candidate to boost performance. With that said, we absolutely agree with the reviewer that approximating the Dmax estimator in other perception-distortion planes is a most interesting topic and we are determined to investigate this in future research.

---

> > ### Comment · Reviewer_YhDf · 2023-08-18
> > **Response to the authors**
> >
> > After reading other reviewers' comments and the rebuttals, I raise my rating.
> >
> > Besides, for the method's effectiveness on real-world datasets, the authors can also utilize some deep learning-based image quality assessment (IQA) methods or user studies for perceptual evaluation.

---

### Official Review · Reviewer_tvbs · 2023-07-10

**Soundness:** 3 good
**Presentation:** 2 fair
**Contribution:** 2 fair
**Rating:** 4
**Confidence:** 4

**Summary:**

In this paper, the authors propose a few-shot algorithm to obtain higher quality restored images of the given model like VAEs and diffusion models. Specifically, the optimal transport map in the latent space is computed through the representations of the real images and the reconstructed images. By applying the OT map, a better restoration can be obtained. Experiments show that the proposed method is effecitive.

**Strengths:**

The paper successfully applying the theory propose by [1] in image restoration and can generate high quality reconstructed images.

**Weaknesses:**

* The presentation of the paper is not good and some concepts are unclear. For example,
    * In line 116-117, what's the meaning of $x^*$ and $\hat{x_0}$
    * Is $x^*$ sampled from $p(x|z)$ that achieves the minimal MSE between the reconstruction and the input? Similarly, how to define $\hat{x_0}$ and what's the meaning of max MSE error?
    * How to define $p_x, p_x^*$?
    * Line 121, OT plan should be between two distributions, not to data points.

* Why does the unpaired dataset give better performance instead of paired dataset? It doesn't seem to make sense and the necessary explanation is needed.

**Questions:**

Please see the weakness part.

---

> ### Author Rebuttal · Authors · 2023-08-06
>
> > In line `116-117`, what's the meaning of $x^*$  and $\hat{x}_0$ ?
>
> $x^*$ is the MMSE estimate, being the posterior mean. This solution gives the best MSE distortion performance, but at the cost of being of poor visual quality. $\hat{x}_0$ on the other hand, is the Dmax solution - being of perfect perceptual quality while giving the smallest distortion possible. Note that these are two extremes in the Perception-Distortion curve, which we refer to in **Figure 1** of the paper. We will update the text in line `116-107` to include these more detailed explanations.
>
> > Is $x^*$  sampled from $p(x|z)$  that achieves the minimal MSE between the reconstruction and the input? Similarly, how to define $\hat{x}_0$ and what's the meaning of max MSE error ?
>
> As $x^*$ is the MMSE, it cannot be obtained as a single sample from the posterior. Note, however, that $x^*$ can be approximated by drawing many samples from $p(x|y)$ and averaging them, as it is the posterior mean. This property is not being used in the paper. As for $\hat{x}_0$, it is defined in line `117` as the estimator attaining the *minimal* MSE while having perfect perceptual quality. This is not to be confused with the *notation* Dmax, referring to this MSE quantitiy, introduced by [(Blau & Michaeli, 2018)](https://openaccess.thecvf.com/content_cvpr_2018/html/Blau_The_Perception-Distortion_Tradeoff_CVPR_2018_paper.html), to refer to the maximal distortion on the perception-distortion curve (see **Figure 1**).
> We will clarify this nuance in the revised version of the paper.
>
> > How to define $p_x$, $p_{x^*}$ ?
>
> $p_x$ and $p_{x^*}$ are the probability distributions of the random variables $x$ and $x^*$. The reviewer is correct to note that the formal definition is lacking in the paper. As we state in **Section 3.2**, $x$ and $x^*$ are random variables defined over $\mathbb{R}^n$, so their definitions of $p_x$ and $p_x^*$ are often omitted for conciseness, like in [(Freirich et. al, 2021)](https://proceedings.neurips.cc/paper/2021/hash/d77e68596c15c53c2a33ad143739902d-Abstract.html). We will add these definitions for completeness.
>
> > Line `121`, OT plan should be between two distributions, not to data points.
>
> The reviewer is correct, and the OT plan is indeed performed between $p_x$ and $p_{x^*}$ as the notation $T_{p_{x^*} \longrightarrow p_x}$ suggests and as we define in `101-102` following [(C. Villani, 2008)](https://cedricvillani.org/sites/dev/files/old_images/2012/08/preprint-1.pdf). We will clarify this better in the revised version of the paper.
>
> > Why does the unpaired dataset give better performance instead of paired dataset? It doesn't seem to make sense and the necessary explanation is needed.
>
> As explained in line `252-253`, it appears that using paired updates actually diverges from the algorithm introduced in **Section 4.1** and might introduce a statistical bias which hinders the covariance matrix estimation. As emphasized in the text, it is only a *hypothesis* and this phenomenon requires further research. Unfortunately, it is not the primary result of our algorithm and its formal explanation is out of the paper’s scope.

---

> > ### Comment · Reviewer_tvbs · 2023-08-21
> >
> > Thanks for the rebuttal. I still think the paper needs to be further polished before publication. Thus, I'll keep my rating.

---

### Decision · Program_Chairs · 2023-09-21

**Decision:**

Accept (poster)

**Comment:**

The final ratings for the rebuttal after the paper were three accepts (two borderline and one weak), with the fourth reviewer recommending reject. However, the negative reviewer did not elaborate on which of their concerns remained and why they were unsatisfied with the rebuttal.

Given the mix of scores, the AC read the paper as well as the reviews and discussions carefully. Based on this reading, the AC believes that the weaknesses listed by the negative reviewer have either been addressed by the rebuttal or are not a bar for acceptance.

Moreover, the AC finds this to be an interesting paper with a novel idea, and with both good theory and practical empirical results. The paper would be of considerable interest to the NeurIPS audience --- beyond offering a new useful algorithm for restoration tasks, the ideas in the paper have the potential to be adapted by other researchers in different settings. Therefore, the AC is happy to recommend acceptance.